# Glycan-dependent cell adhesion mechanism of Tc toxins

Daniel Roderer [1], Felix Bröcker[2,3], Oleg Sitsel [1], Paulina Kaplonek[2,4], Franziska Leidreiter[1,5], Peter H. Seeberger[2] & Stefan Raunser [1✉]

Toxin complex (Tc) toxins are virulence factors of pathogenic bacteria. Tcs are composed of three subunits: TcA, TcB and TcC. TcA facilitates receptor–toxin interaction and membrane permeation, TcB and TcC form a toxin-encapsulating cocoon. While the mechanisms of holotoxin assembly and pore formation have been described, little is known about receptor binding of TcAs. Here, we identify heparins/heparan sulfates and Lewis antigens as receptors for different TcAs from insect and human pathogens. Glycan array screening reveals that all tested TcAs bind negatively charged heparins. Cryo-EM structures of *Morganella morganii* TcdA4 and *Xenorhabdus nematophila* XptA1 reveal that heparins/heparan sulfates unexpectedly bind to different regions of the shell domain, including receptor-binding domains. In addition, *Photorhabdus luminescens* TcdA1 binds to Lewis antigens with micromolar affinity. Here, the glycan interacts with the receptor-binding domain D of the toxin. Our results suggest a glycan dependent association mechanism of Tc toxins on the host cell surface.

[1] Department of Structural Biochemistry, Max Planck Institute of Molecular Physiology, 44227 Dortmund, Germany. [2] Department of Biomolecular Systems, Max Planck Institute of Colloids and Interfaces, 14476 Potsdam, Germany. [3]Present address: Vaxxilon Deutschland GmbH, 12489 Berlin, Germany. [4]Present address: Ragon Institute of MGH, MIT and Harvard, Cambridge, MA 02139, USA. [5]Present address: Department of Biomolecular Mechanisms, Max Planck Institute for Medical Research, 69120 Heidelberg, Germany. ✉email: stefan.raunser@mpi-dortmund.mpg.de

A widespread type of toxin in insect and human bacterial pathogens is the heterotrimeric toxin complex (Tc)[1,2]. Tc toxins were originally discovered in the insect pathogen *Photorhabdus luminescens*, which lives in symbiosis with soil nematodes[3]. Later, gene loci of Tc toxins were also found, amongst others, in *Xenorhabdus nematophila*[4,5], in the facultative human pathogen *Photorhabdus asymbiotica*[6], and different species of *Yersinia*, which are both human and insect pathogens[7,8].

Tc toxins consist of three components: TcA, TcB, and TcC. The ~1.4 MDa TcA is a homo-pentameric bell-shaped molecule that mediates target cell association, membrane penetration, and toxin translocation[9]. TcA is made up of a central, pre-formed channel that is enclosed by an outer shell composed of a structurally conserved α-helical domain decorated with variable receptor-binding domains (RBDs). The shell and channel are connected by a stretched linker[10]. TcB and TcC together form a ~250 kDa cocoon that encapsulates the toxic component, the ~30 kDa C-terminal hypervariable region of TcC, which is autoproteolytically cleaved and resides inside the cocoon[10,11].

Binding of TcB–TcC to TcA involves a conformational transition of the TcA-binding domain of TcB, which is a distorted six-bladed β-propeller that closes the cocoon at the bottom. Following the contact of TcA and TcB, two blades of the β-propeller unfold and refold, resulting in the opening of the cocoon. As a consequence, the toxic enzyme enters the translocation channel of TcA after holotoxin assembly[12].

After specific binding to receptors on the surface of the target cell, the holotoxin is endocytosed[13]. Acidification in the late endosome triggers the opening of the shell, which results in the compaction of the stretched linker that drives the channel through the now open shell into the target cell membrane. The tip of the channel then opens in the membrane and enables translocation of the actual toxin[14,15].

While the mechanisms of holotoxin assembly and prepore-to-pore transition have been well described (reviewed in ref. [16]), little is known regarding the receptor binding and cellular uptake of Tc toxins. In the case of Yen–TcA, the TcA from the insect pathogen *Yersinia entomophaga*, two chitinases form a complex with the toxin[17]. It was proposed that the chitinases cause the degradation of the peritrophic membrane within the midgut of insects to facilitate toxin entry[18]. The association of Yen–TcA with target cells potentially occurs via glycan structures, which were recently identified in a binding screen to associate with both chitinases and directly with Yen-Tc[19].

Various other toxins have been shown to bind to glycans on target cells, such as botulinum neurotoxin, which binds to N-glycosylated SV2 (ref. [20]), cholera toxin, which binds to ganglioside GM1 (ref. [21]), and typhoid toxin from *Salmonella typhi*, which binds to multiantennal glycans[22]. The structural similarity of one of the shell-enclosing RBDs of various TcAs to sialidase[10,23] suggests that glycans could be also the cellular receptors in the case of TcAs that do not associate with chitinases.

Here, we use a synthetic glycan microarray[24] to specifically screen for glycans as possible receptors for Tc toxins from several insect and human pathogens (*P. luminescens* TcdA1 (Pl-TcdA1), *X. nematophila* XptA1 (Xn-XptA1), *Morganella morganii* TcdA4 (Mm-TcdA4), *Yersinia pseudotuberculosis* TcaATcaB (Yp-TcaATcaB)). All tested TcAs interact with heparins/heparan sulfates (HS). Cryo-EM structures of Mm-TcdA4 and Xn-XptA1 in complex with heparin reveal that the glycan interacts with different parts of the shell domain, including receptor-binding domains. In addition, Pl-TcdA1 interacts with several Lewis antigens, which bind to the RBD D as revealed by the cryo-EM structure of the Pl-TcdA1 in complex with BSA-Lewis X.

## Results

### Complex glycans on the cell surface mediate TcA association.

When intoxicating different cell types with Pl-TcdA1, we found that HEK 293 GnTI⁻ cells are less susceptible to the toxin than HEK 293T cells (Supplementary Fig. 1a, b). While HEK 293T cells readily bind and accumulate fluorescently labeled Pl-TcdA1 after 1–4 h of incubation time, HEK 293 GnTI⁻ cells bind considerably less fluorescent Pl-TcdA1 (Fig. 1a, b). The major difference between the two cell types is that HEK 293 GnTI⁻ cells do not have N-acetyl-glucosaminyltransferase I and therefore lack complex N-glycans[25]. The lack of mature N-linked glycans on the plasma membrane of HEK 293 GnTI⁻ cells is probably responsible for the decreased interaction with Pl-TcdA1, suggesting that N-linked glycans play a major role in the specific toxin binding to the host membrane.

To confirm that this is indeed the case, we treated HEK 293T cells with PNGase F to completely remove N-linked glycans on the plasma membrane[26]. Pl-TcdA1 binding to these cells was tremendously decreased supporting our conclusion that N-linked glycans are important for the binding of the toxin to host cells (Fig. 1c and Supplementary Fig. 1b).

### Pl-TcdA1 binds to Lewis antigens.

In order to identify possible glycans that are responsible for the specific binding of Tc toxins to host cells, we first applied a synthetic glycan microarray and screened TcAs from different organisms, namely Pl-TcdA1, Xn-XptA1, Mm-TcdA4, and Yp-TcaATcaB. We identified several Lewis oligosaccharides that interacted with Pl-TcdA1 (Fig. 2a, b). Especially the trisaccharide Lewis X and the tetrasaccharides Lewis Y and sialyl-Lewis X interacted strongly with Pl-TcdA1 (Fig. 2b, c and Supplementary Fig. 2a). Interestingly, none of the other TcAs showed a significant binding to the microarray even at higher protein concentrations, indicating that this glycan–toxin interaction is specific for Pl-TcdA1.

The interaction with Lewis oligosaccharides is unexpected since these α1,3-fucosylated glycans are normally found on the surface of human red blood cells[27] and not in insects. Therefore, it is unlikely that they represent the natural receptor of Pl-TcdA1, which is a Tc toxin from an insect pathogenic bacterium. Although Lewis antigens have not been described for insects, similar oligosaccharides with α1,3-fucosylated Lewis-like antennae have been reported to be present in glycoproteins of the lepidopterans *Trichoplusia ni* and *Lymantria dispar*[28]. Interestingly, *Lepidoptera* are typical hosts of nematodes living in symbiosis with *P. luminescens*[3] and are therefore the natural target of Tc toxins. We therefore believe that the interaction between Pl-TcdA1 and Lewis antigens is representative for the binding of the toxin to similar glycan receptors on the insect host membrane.

Protein–glycan interactions are generally weak and multivalent interactions are in many cases necessary to achieve a significant increase in receptor binding affinity[29]. Since TcA is a homo-pentameric complex, high-affinity receptor binding is likely also achieved by multivalency. To determine the binding kinetics of Lewis X antigens (compound 7 in Fig. 2c) to TcA pentamers, we conjugated multiple oligosaccharides to BSA used as a carrier protein (Supplementary Fig. 2b–d) and quantified the protein–oligosaccharide interaction using biolayer interferometry (BLI). This approach mimics the high surface density of glycans on a cell surface and thereby ensures that all Lewis X binding sites on a TcA pentamer can be occupied.

We determined a dissociation constant ($K_D$) of $5.8 \pm 0.7 \, \mu M$ (Fig. 2d) for the complex of Pl-TcdA1 and the BSA-Lewis X glycoconjugate. The relatively high on- and off-rates of $3.8 \pm 0.4 \times 10^4 \, M^{-1} \, s^{-1}$ and $0.22 \pm 0.004 \, s^{-1}$, respectively, indicate a highly

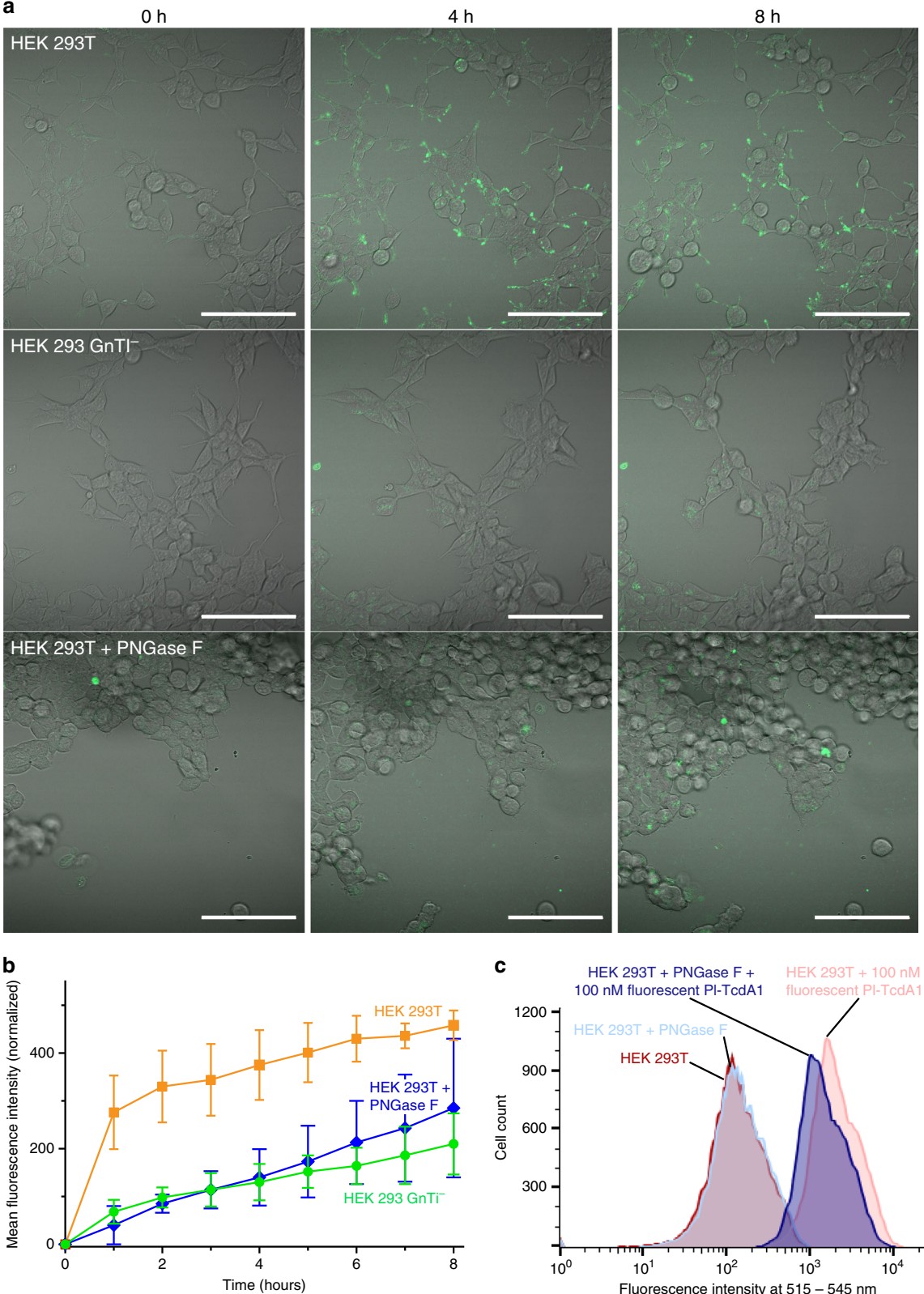

**Fig. 1 The presence of complex glycans on target cells enhances the binding of Pl-TcdA1. a** Time course confocal imaging demonstrates that wild type HEK 293T cells bind Pl-TcdA1 (AlexaFluor488 labeled, green) much more readily than their glycosylation-deficient HEK 293 GnTI⁻ counterparts and PNGase F-treated HEK 293T cells. Images were taken after 0, 4, or 8 h of incubation at 37 °C. Scale bar, 100 μm; fluorescence channels are equally thresholded. **b** Quantification of AlexaFluor488 labeled Pl-TcdA1 binding to wild type HEK 293T (orange squares), PNGase F-treated HEK 293T (blue diamonds), and HEK 293 GnTI⁻ cells (green circles) based on time course confocal microscopy data. Three biological replicates for each cell type were analyzed; error bars represent standard deviation. **c** Flow cytometry of HEK 293T exposed to AlexaFluor488 labeled Pl-TcdA1 after deglycosylation of cell surface by PNGase F. Histograms of PNGase F-treated and untreated cells, both with and without 100 nM Pl-TcdA1 are shown in comparison. The source data underlying panels (**b**) and (**c**) are provided as a Source Data file.

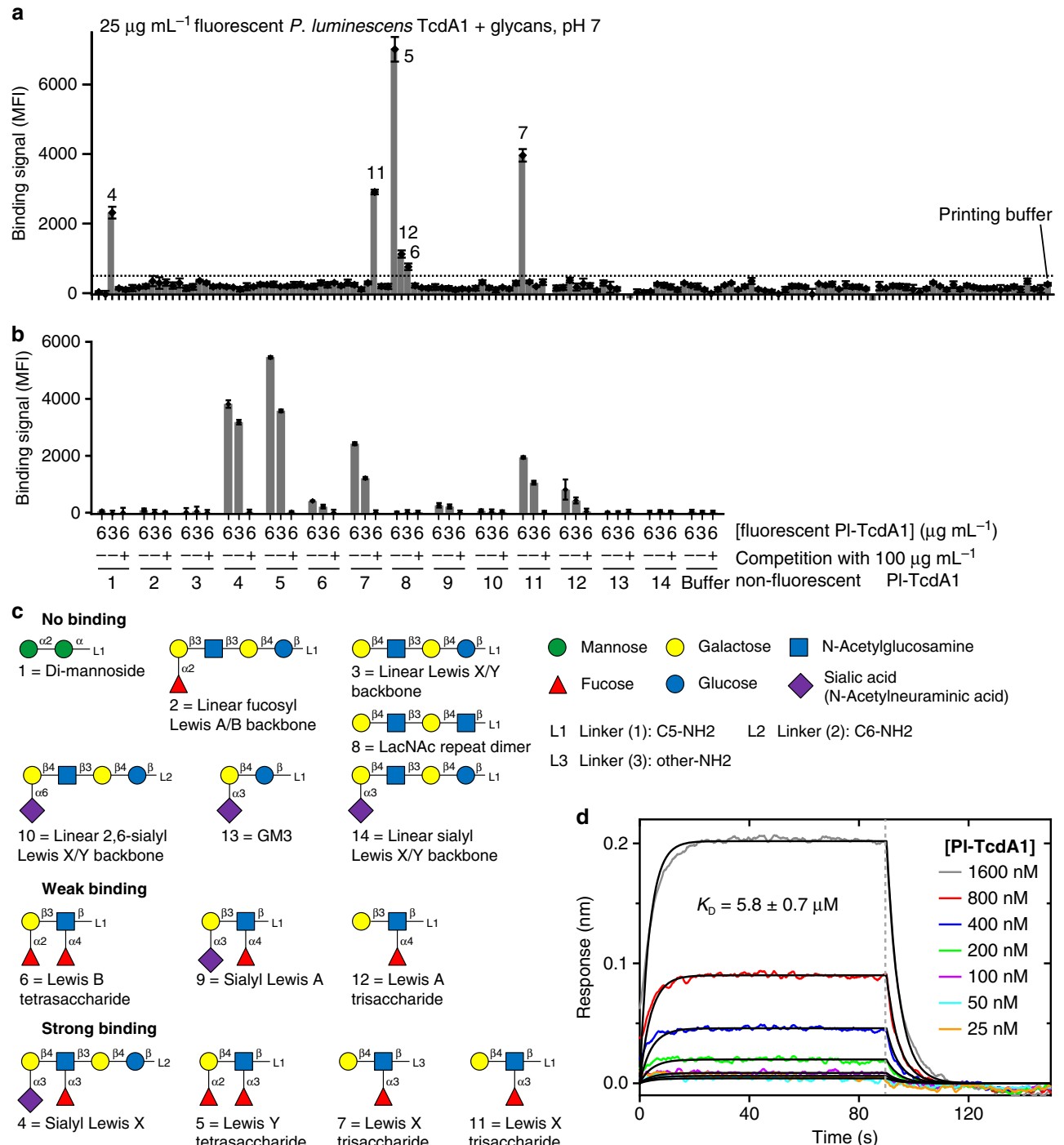

**Fig. 2 Interaction of Pl-TcdA1 with various Lewis antigens. a** Glycan microarray showing the interaction of fluorescently labeled Pl-TcdA1 with various glycans. Binding signals are shown as local background-subtracted mean fluorescence intensity (MFI) values to 141 synthetic glycans. MFI values are indicated as bars. An arbitrary threshold of MFI = 500 is indicated as dashed line. **b** Focused microarray with selected glycans from (**a**). 6 or 3 μg/mL of fluorescent Pl-TcdA1 were applied to 14 different glycans (indicated below the bar diagrams, spotted at 0.1 mM) in the absence or presence of an excess of unlabeled Pl-TcdA1. The error bars in panels (**a**) and (**b**) represent standard deviations of four measurements. **c** Schematic representation of glycans used on the focused microarray, grouped by binding of Pl-TcdA1 (no binding, weak binding, and strong binding). The individual monosaccharide moieties are shown on the top right (green circle: mannose, yellow circle: galactose, blue square: *N*-acetylglucosamine, red triangle: fucose, blue circle: glucose, violet diamond: sialic acid). For subsequent cryo-EM and BLI experiments, Lewis X (compound 7) was conjugated to BSA (Supplementary Fig. 2b). **d** BLI sensorgrams of Pl-TcdA1 interacting with immobilized BSA-Lewis X. TcA pentamer concentrations were 25–1600 nM. A global fit according to a 1:1 binding model was applied (black curves), resulting in a dissociation constant ($K_D$) of 5.8 ± 0.7 μM, $k_{on}$ of 3.8 ± 0.4 × 10$^4$ M$^{-1}$s$^{-1}$, and $k_{off}$ of 2.2 ± 0.04 × 10$^{-1}$s$^{-1}$. Association and dissociation phases are separated by a gray dashed line. The source data underlying panels (**a**) and (**b**) are provided as a Source Data file.

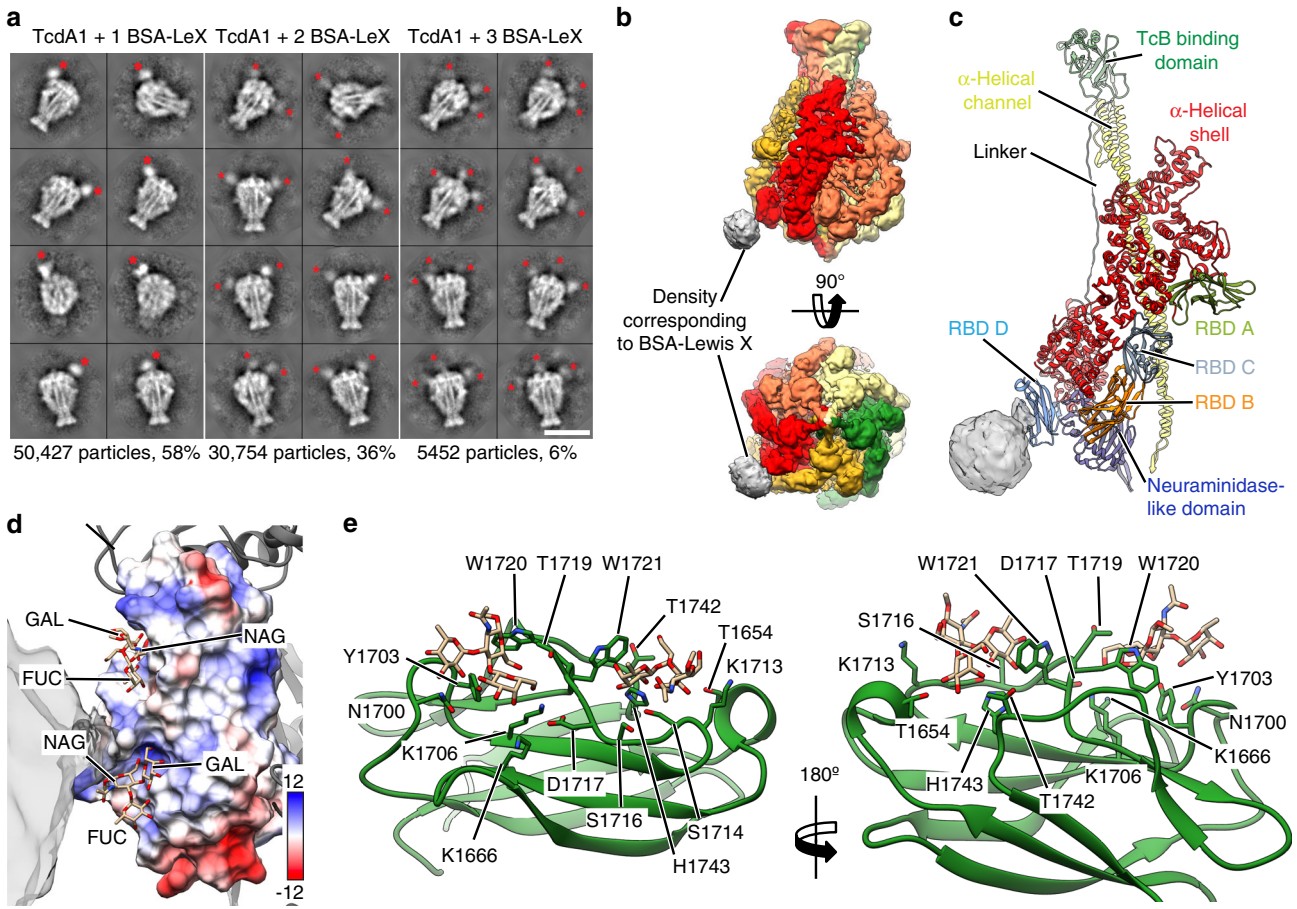

**Fig. 3 Structure of Pl-TcdA1 with BSA-Lewis X. a** Representative reference-free 2D class averages of cryo-EM images of Pl-TcdA1 with crosslinked BSA-Lewis X obtained by ISAC and resampled to the original pixel size. Scale bar, 20 nm. **b** Cryo-EM map of Pl-TcdA1 with BSA-Lewis X. The protomers are colored individually. Density corresponding to BSA-Lewis X (gray) is shown at lower binarization threshold. **c** Model of one protomer of Pl-TcdA1 together with the cryo-EM density of BSA-Lewis X (gray) at low binarization threshold. The domains of Pl-TcdA1 are colored individually. BSA-Lewis X is attached to receptor binding domain D (RBD D). **d** Surface representation of RBD D, colored according to the Coulomb potential (kcal mol$^{-1}$ e$^{-1}$) at pH 7.0. Two docked conformations of Lewis X are shown as stick representations. The docking score is −4.5 kcal mol$^{-1}$ for both conformations. The cryo-EM density map corresponding to BSA-Lewis X is transparent gray. FUC: fucose, GAL: galactose, NAG: N-acetylglucosamine. **e** Detailed view of RBD D (green) with two docked Lewis X molecules. Sidechains that contact the docked molecules are highlighted.

dynamic binding equilibrium between Lewis X and Pl-TcdA1. The complex of Pl-TcdA1 with its as yet unknown natural glycosylated protein receptor is probably more stable, since the peptide moiety of the receptor might also contribute to binding.

**Structure of the Pl-TcdA1–BSA-Lewis X complex.** To elucidate how Lewis X interacts with Pl-TcdA1, we planned to determine the structure of the Pl-TcdA1–BSA-Lewis X complex by single particle cryo-EM. However, since the affinity of Pl-TcdA1 to BSA-Lewis X is low, high concentrations of BSA-Lewis X are required to allow for complex formation which is incompatible with single particle cryo-EM. Therefore, we stabilized the Pl-TcdA1–BSA-Lewis X complexes that were formed using a large excess of BSA-Lewis X by crosslinking with glutaraldehyde and removed unbound BSA-Lewis X. The resulting complexes were suitable for single particle cryo-EM (Supplementary Fig. 3a). Analyzing the single particles by two-dimensional clustering and sorting in SPHIRE[30] revealed up to three additional densities corresponding to BSA-Lewis X at the periphery of Pl-TcdA1, indicating that the glycan interacts with one of the RBDs of the toxin (Fig. 3a and Supplementary Fig. 3b). Since not all five Lewis X binding sites of the Pl-TcdA1 pentamer are occupied and the

number of BSA-Lewis X molecules per complex varies, we combined three-dimensional classification with symmetry expansion (see "Methods" and Supplementary Fig. 4) to obtain a three-dimensional reconstruction of the Pl-TcdA1–BSA-Lewis X complex, in which one BSA-Lewis X is bound to one Pl-TcdA1 pentamer (Fig. 3b).

Despite the heterogeneity of the data set and the flexibility of the bound BSA-Lewis X complex, the reconstruction reached an average resolution of 5 Å (Supplementary Fig. 3c–e). This enabled us to accurately fit the previously determined cryo-EM structure of Pl-TcdA1 into the toxin density[23] (Fig. 3c). Although the density corresponding to BSA-Lewis X is less well resolved and therefore the glycan or BSA cannot be fitted, its position on the toxin unambiguously demonstrates that it binds to RBD D (Fig. 3b, c and Supplementary Movie 1).

Molecular docking of the Lewis X trisaccharide on RBD D revealed two prominent potential binding sites (Fig. 3d, e and Supplementary Movie 1). Separated by two tryptophan residues (W1720 and W1721), both binding sites form positively charged grooves with many polar residues (predominantly Thr, Ser, and Lys), allowing hydrogen bond mediated interactions with the glycan.

**TcAs of insect and human pathogens bind to heparins**. A specific type of glycosaminoglycans, heparan sulfate (HS), has been shown to function as a receptor for viruses, such as herpes simplex virus[31], dengue virus[32], and adeno-associated virus[33]. Heparin is primarily expressed in mast cells, while HS is ubiquitous on cell surfaces and in the extracellular matrix. Heparins and HS are considered to be the most structurally complex glycosaminoglycans[34], being composed of disaccharide repeating units of D-glucosamine (GlcN) and either L-iduronic acid (IdoA) or D-glucuronic acid (GlcA) linked by α1-4 or β1-4 glycosidic bonds. Sulfation can occur at positions 2, 3, and 6 of GlcN as well as at position 2 of IdoA/GlcA. In heparin, uronic acid residues are 90% IdoA and 10% its C5 epimer GlcA. The prototypical heparin disaccharide contains three sulfate groups (2.7 sulfates per disaccharide on average in the polymer). HS chains are typically more heterogeneous than those of heparin, are richer in N-acetyl D-glucosamine (GlcNAc) and GlcA and contain fewer O-sulfates (on average one per disaccharide in the polymer).

Since one of the surface domains of TcAs is structurally homologous to virus neuraminidases[10], we decided to assess whether also heparins/HS interact with the four different TcAs Pl-TcdA1, Xn-XptA1, Mm-TcdA4, and Yp-TcaATcaB. Therefore, we performed a glycan screen comprising different heparins/HS oligosaccharides. We found that all tested TcAs interact with heparin or HS (Fig. 4a and Supplementary Fig. 5a, b). The binding signals that were obtained for Mm-TcdA4 and Xn-XptA1 were in the same range as those obtained for adenovirus-2, which had been screened on the same array layout[33]. The binding signals of Pl-TcdA1 and Yp-TcaATcaB were weaker and reached up to 40% of the values of Mm-TcdA4 for identical heparins. The controls (wheat germ agglutinin and BSA) showed almost no binding to the microarray (Fig. 4a), indicating that the heparin/HS-toxin interaction is indeed specific. While the chain length of the strongly binding heparins/HSs varies and seems therefore not to be crucial for the interaction (Fig. 4b), we observed that all of them had at least two negative charges per monosaccharide group, suggesting that the binding of TcA to heparins/HS is facilitated by electrostatic interactions.

**Structure of the Mm-TcdA4–heparin complex**. We chose Mm-TcdA4 from the opportunistic human pathogen *M. morganii* to investigate the heparin/HS-toxin interaction in more detail. First, we performed BLI to determine the affinity of Mm-TcdA4 to heparin/HS. Since synthetic heparins/HSs are difficult to produce in the large amounts needed for biophysical and structural studies, we decided to work with natural ~15 kDa porcine intestinal mucosa heparin instead, regardless of its heterogeneity. We obtained BLI curves that show a rapid, concentration-dependent signal increase upon association and a signal decrease upon dissociation, indicating the tight interaction of Mm-TcdA4 and heparin (Supplementary Fig. 5c). The heterogeneity of the natural heparin made it impossible to determine the exact $K_D$, but the comparison with the BLI curves of Pl-TcdA1 and BSA-Lewis X (Fig. 2d) indicate an affinity that is at least 1 order of magnitude higher due to the slower dissociation.

To identify which domains of Mm-TcdA4 are involved in heparin/HS binding, we determined the structure of Mm-TcdA4 in complex with porcine intestinal mucosa heparin using single particle cryo-EM (Supplementary Fig. 6 and Supplementary Movie 2). The 3.2 Å structure is very similar to the previously obtained structure of Mm-TcdA4 (ref. [23]) (Supplementary Fig. 6g). However, we identified additional densities at the periphery of the shell domain of each protomer (Fig. 4c, Supplementary Fig. 8a, b, and Supplementary Movie 2). The density occupies a cleft that is predominantly formed by

positively charged residues (R194, R195, K243, K265, R266, and R886), representing an ideal binding site for negatively charged heparins (Fig. 4d). This shows that unexpectedly heparins do not interact with an RBD or the neuraminidase-like domain at the tip of the shell, but directly with the α-helical domain of the shell. Although the resolution of the density does not allow to build an atomic model in this region, we flexibly fitted several heparin oligosaccharides that resemble natural heparin into the density. Indeed, most of them fit well and their negatively charged residues are involved in many potential electrostatic interactions at the interface (Fig. 4e and Supplementary Fig. 8c).

**Structure of the Xn-XptA1–heparin complex**. Since the surface of the α-helical shell is one of the least conserved regions of TcA[23], the positively charged heparin/HS binding site of Mm-TcdA4 is not present at the same position in other Tc toxins. To address whether heparins/HS bind to a different region in other TcAs, we decided to determine the structure of a TcA from another organism in complex with heparin/HS. We chose Xn-XptA1, because it also shows a strong binding signal to porcine intestinal mucosa heparin (Fig. 4a). Expectedly, the obtained cryo-EM structure at 3.7 Å resolution is very similar to the structure of Xn-XptA1 without heparin[23] (Supplementary Fig. 7), but clear densities corresponding to heparin/HS are present (Fig. 4f and Supplementary Movie 3). However, their position differs from that in the Mm-TcdA4–heparin complex. Instead of interacting with the upper region of the shell, they are located in a gap between the neuraminidase-like domain, RBD B and RBD D (Fig. 4f, g, Supplementary Fig. 8d, e, and Supplementary Movie 3). This binding pocket is not directly connected to the α-helical shell and is predominantly formed by polar and charged residues. In particular, a docked heparin oligosaccharide interacts with N1090, N1092, H1121, D1582, and S1584 of the neuraminidase-like domain, Y1548 and S1580 of RBD B and W1663, K1665, N1674, R1677, Q1678, and P1679 of RBD D (Fig. 4h and Supplementary Fig. 8f). Since the RBDs of Tc toxins are not conserved, this binding pocket is not conserved as well.

Taken together, our results show that the heparin/HS binding site is not conserved in different TcAs, although binding itself is. The non-conserved binding of different TcAs to heparin/HS indicates that these glycosaminoglycans function as co-receptors that concentrate Tc toxins on the host cell surface, increasing the toxins' ability to bind specifically to secondary receptors in further steps.

## Discussion

In the current work, we demonstrate that Tc toxins specifically interact with glycans on the surface of host cells. We identified two specific types of glycans that act as receptors for Tc toxin binding: heparins/HSs and Lewis antigens. While heparins bind to all tested TcAs, Lewis antigens only interact with Pl-TcdA1, specifically its RBD D.

To understand why Lewis antigens only bind to the RBD D of Pl-TcdA1 and not to that of other TcAs, we performed a structure-based sequence alignment of the RBD Ds of four TcAs with known structures (Pl-TcdA1, Xn-XptA1, Mm-TcdA4, and Pl-TcdA4). The alignment reveals that although structurally similar binding pockets exist in all RBD Ds, they are not conserved. This also includes the residues W1720 and W1721 that separate the pockets in Pl-TcdA1 (Supplementary Fig. 9a–c), suggesting that the RBDs from different Tc toxins interact with different receptors. This is in line with our experimental results, showing that binding of Lewis X is specific for the RBD D of Pl-TcdA1.

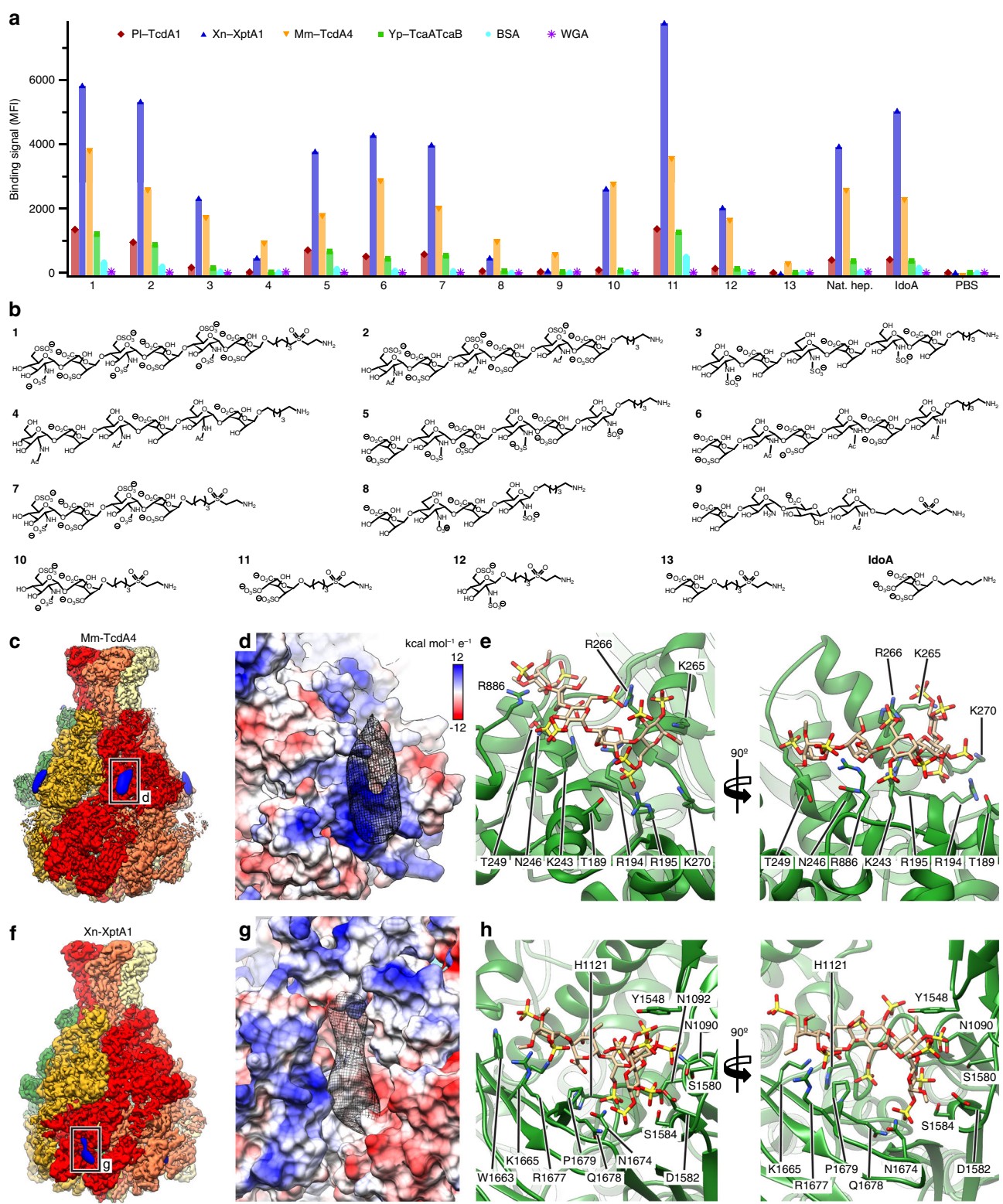

It has also been reported that Lewis X binds to cholera toxin[35]. As in the case of Pl-TcdA1, the Lewis X binding site of cholera toxin is located in a groove. Interestingly, the binding pocket contains a threonine, an asparagine and two histidine residues[35] and therefore strongly resembles one of the Lewis X binding pockets of Pl-TcdA1 (Supplementary Fig. 9d, e).

The heparin/HS binding site in the α-helical shell domain of Mm-TcdA4 is 10 nm away from the membrane if the Tc toxin is in the prepore conformation and attaches perpendicularly to the membrane. In the case of Xn-XptA1, the heparin/HS binding site is closer to the membrane (ca. 4 nm) and is located in a cleft between two RBDs and the neuraminidase-like domain. HS proteoglycans form long polymers on cell surfaces that extend 20–150 nm from the plasma membrane[36,37]. Thus, they can easily bind to the toxin even before it reaches the membrane surface. We propose that Tc toxins bind first to HS, which brings them closer to the cell surface and thereby into contact with other surface receptor molecules. This would be reminiscent of how HS

**Fig. 4 Interaction of TcAs with various heparin oligosaccharides and structures of Mm-TcdA4 and Xn-XptA1 in complex with heparin. a** Glycan microarray showing the interaction of the indicated TcAs (25 µg mL$^{-1}$) with selected heparin oligosaccharides, the natural heparin mixture from porcine intestinal mucosa (Nat. hep.), and the heparin analogue IdoA-2,4-disulfate(a)linker (IdoA). Bovine serum albumin (BSA) and wheat germ agglutinin (WGA) were used as negative controls. Phosphate-buffered saline (PBS) was spotted on the array as an additional negative control. The full heparin microarray is shown in Supplementary Fig. 5a. **b** Structures of heparins 1–13 and IdoA. The numbers correspond to (**a**). **c** Cryo-EM structure of Mm-TcdA4 in complex with natural heparin. The protomers of Mm-TcdA4 are colored individually. Additional density not corresponding to the toxin is filtered to 22 Å and shown in blue. **d** Surface representation of the heparin binding site of the shell domain of Mm-TcdA4, colored according to the Coulomb potential (kcal mol$^{-1}$ e$^{-1}$) at pH 7.0. The additional density corresponding to heparin is shown in black mesh representation. **e** The interface region of the Mm-TcdA4 shell domain (green) with sidechains that contact a docked heparin pentasaccharide (sand colored). The docking score is −6.2 kcal mol$^{-1}$. All residues belong to the α-helical part of the shell. **f** Cryo-EM structure of Xn-XptA1 in complex with natural heparin. The protomers of Xn-XptA1 are colored individually. Additional density not corresponding to the toxin is filtered to 22 Å and shown in blue. **g** Surface representation of the heparin binding site of the shell domain of Xn-XptA1, colored according to the Coulomb potential analogous to (**d**). The additional density corresponding to heparin is shown in black mesh representation. **h** The interface region of Xn-XptA1 with sidechains that contact a docked heparin pentasaccharide (sand colored). The docking score is −6.1 kcal mol$^{-1}$. The residues forming the heparin binding site are located in RBD B, the neuraminidase-like domain and RBD D. The source data underlying panel (**a**) are provided as a Source Data file.

binds to viruses (e.g., herpes virus[38], baculovirus[39], and adeno-associated virus[33]) and cationic cell-penetrating peptides[40], acting as a receptor for their cell entry. Usage of the heparan sulfate carrying protein syndecan-1 as a cellular receptor by *Staphylococcus aureus* beta toxin represents an interesting variation of this scenario[41].

On the other hand, Lewis antigens are shorter than glycosaminoglycans, meaning that the toxin has to be closer to the membrane in order to interact with them. We therefore propose that Pl-TcdA1 binds to Lewis antigens as a second step. Because the five identical RBD Ds which interact with Lewis antigens are symmetrically arranged at the bottom of the Pl-TcdA1 shell, the binding affinity is highest when the toxin is oriented perpendicularly to the membrane. This puts the toxin into the ideal position for membrane penetration.

At first glance, the low affinity of Pl-TcdA1 to BSA-Lewis X contradicts such a tight association. However, the artificial BSA-Lewis X receptor lacks both the native arrangement of Lewis X and the correct protein moiety of the native glycoprotein receptor of Pl-TcdA1. The presence of both probably provide a higher overall affinity, therefore the identification of the as yet unknown native receptor and the co-structure of the receptor and Pl-TcdA1 will provide further insights into the cell association mechanism of Tc toxins.

Lewis antigens are normally found on the surface of human red blood cells[27] and have been identified as receptors for toxins of human pathogens and viruses, such as cholera toxin[35] or norovirus[42]. However, Lewis-like oligosaccharides with core α1,3-fucosylation also have been identified in various insect species[28,43], making them likely receptors for Tc toxins from entomopathogenic bacteria. Conversely, the conserved Lewis X core structure is one of the receptor motifs that explains the high toxicity of Pl-TcdA1 against various mammalian cells, although insects of the order *Lepidoptera* are the naturally targeted organisms.

In a previous glycan array screen with the Tc toxin Yen-Tc from the insect pathogen *Y. entomophaga*, 72 out of 432 glycans were identified as potential toxin receptors, amongst them monosaccharides, fucosylated glycans, mannose derivatives, glucose derivatives, sialylated derivatives, glycosaminoglycans, complex *N*-glycans and derivatives with terminal GlcNAc, GalNAc, or galactose[19]. In contrast, Xn-XptA1, Mm-TcdA4, and Yp-TcaATcaB only interacted strongly with heparins/HS (11 out of 36 heparins/HS on the microarray). Pl-TcdA1 additionally bound to core-fucosylated Lewis antigens (6 out of 141 glycans on the microarray). This indicates that the chitinase-associated Yen-Tc is clearly less selective in terms of glycan binding than the TcAs that we tested here.

What about the other RBDs? Their structure is similar to RBD D, suggesting that they may also interact with glycans that we have not identified so far. There are three possible explanations why these interactions may have not been detected by our microarray screen. First, the distance between the glycans on the array surface and the RBDs in the toxins differ from the situation on cell surfaces, where the glycans are mostly bound to proteins. Therefore, possible glycan binding especially to RBDs that locate distally to the membrane might be undetectable, especially when considering the effect of multivalency in the pentameric TcA. Second, our glycan screen covers only a selection of all glycans that exist on target cells. For example multi-antennal glycans, which have been described to function as high-affinity receptors for typhoid toxin[22], were not present in the array. Third, it could well be that several of the RBDs do not bind to glycans on the cell surface, but directly to proteins.

The removal of N-linked glycans on HEK 293T cells or the absence of complex glycans on HEK 293 GnTI$^-$ cells reduced the number of Pl-TcdA1 toxins bound to the cell surface when compared to untreated HEK 293T cells, but it did not completely prevent toxin binding. This indicates that the toxin does not only bind to N-linked glycans, but probably also interacts with O-linked glycans and/or directly with proteins or lipids in the host cell membrane. However, the decreased toxicity seen in N-linked glycan deficient cells suggests that only synergistic binding will provide the affinity needed to infect cells with high specificity at sub-nanomolar Tc toxin concentrations.

Taken together, our data allow us to propose the following model for Tc toxin binding to cells (Fig. 5). Initially, Tc toxins bind to long heparin/heparan sulfate glycosaminoglycans, which bring them into close contact to other surface receptor molecules. Thereupon, the toxin interacts with these receptors to orient itself into an ideal position for membrane penetration. It then undergoes endocytosis, and subsequent acidification in the late endosome triggers the TcA prepore-to-pore transition. This results in membrane permeation, and allows the release of the cytotoxic cargo that eventually kills the cell.

## Methods

**Protein production.** *P. luminescens* TcdB2–TccC3 was expressed in *Escherichia coli* BL21-CodonPlus(DE3)-RIPL using pET28a (Novagen) with a fusion construct of the genes coding for Pl-TcdB2 and Pl-TccC3 and an N-terminal His$_6$-tag[12]. 10 L LB medium were inoculated with a fresh transformant and protein expression was directly started with 27 µM isopropyl-β-D-thiogalactopyranoside (IPTG). Expression was carried out for 4 h at 28 °C, followed by 20 h at 25 °C and 24 h at 20 °C. Subsequently, the cells were lysed in 20 mM Tris–HCl (pH 8.0), 300 mM NaCl, 10% glycerol (lysis buffer) using a microfluidizer. Cell debris was centrifuged down for 30 min at 35,000 rpm in a Beckman TI45 rotor at 4 °C and the supernatant was loaded on a 5 mL HisTrap FF column (GE Healthcare Life Sciences) equilibrated in lysis buffer. After first washing with 30 mL lysis buffer and then with 30 mL lysis

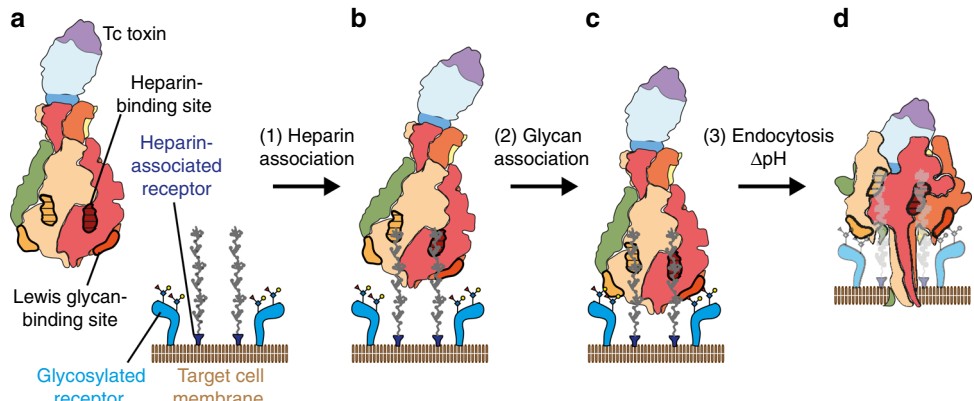

**Fig. 5 Model of the intoxication mechanism of Tc. a** Tc toxin and target cells with multiple glycosylated and heparin-associated receptors. The binding sites for heparin and Lewis antigens are indicted on two protomers. **b** Tc toxin associates to heparin-associated receptors on target cells. **c** In the second association reaction, Tc toxin binds to additional glycosylated receptors, resulting in a tighter toxin-cell complex. **d** Upon endocytosis, the shift of pH to acidic values induces prepore-to-pore transition and membrane permeation of the Tc toxin.

buffer plus 20 mM imidazole, the protein was eluted with a linear gradient from 20 to 300 mM imidazole over 200 mL. Protein-containing fractions were pooled, diluted 16-fold with 20 mM Tris–HCl (pH 8.0), 5% glycerol (buffer A) and loaded on a 5 mL HiTrap Q FF column (GE Healthcare Life Sciences) equilibrated in buffer A. After washing with 30 mL buffer A, Pl-TcdB2-TccC3 was eluted with a linear gradient from 0 to 600 mM NaCl over 200 mL. The protein-containing fractions were pooled and loaded on a Superdex 200 16-60 column equilibrated in 20 mM Tris–HCl (pH 8.0), 150 mM NaCl, 5% glycerol. Purified Pl-TcdB2-TccC3 was either directly used or supplemented with 20% glycerol and stored at −80 °C.

The four TcAs (Pl-TcdA1, Xn-XptA1, Mm-TcdA4, and Yp-TcaATcaB) were expressed in *E. coli* BL21-CodonPlus(DE3)-RIPL using pET19d (Novagen) with the genes coding for Pl-TcdA1, Xn-XptA1, Mm-TcdA4, or pET28a with the genes coding for Yp-TcaA and Yp-TcaB as a fusion construct, resulting in Yp-TcaATcaB[23]. All proteins were modified with an N-terminal His₆-tag. Cells were transformed with the respective plasmid and grown in 10 L medium (Xn-XptA1, Pl-TcdA1, and Yp-TcaATcaB in LB medium and Mm-TcdA4 in 2TY medium) at 37 °C. At an optical density at 600 nm (OD$_{600}$) of 0.8, protein expression was induced with 25 μM IPTG and the expression was carried out at 20 °C for 20 h (Pl-TcdA1, Mm-TcdA4) or 18 °C for 20 h (Xn-XptA1 and Yp-TcaATcaB). Cells were lysed in 50 mM Tris–HCl (pH 8.0), 300 mM NaCl, 0.05% Tween 20 (Pl-TcdA1, Mm-TcdA4) or 20 mM Tris–HCl (pH 8.0), 150 mM NaCl, 0.5% Triton X-100 (Xn-XptA1, Yp-TcaATcaB) using a microfluidizer. All lysis buffers were supplemented with 200 μM Pefabloc. Cell debris was centrifuged down for 30 min at 38,000 rpm in a Beckman TI45 rotor at 4 °C and the supernatants were loaded on a 5 mL HisTrap FF column (GE Healthcare Life Sciences) equilibrated in the respective lysis buffer. After first washing with 30 mL lysis buffer and then with 30 mL lysis buffer plus 20 mM imidazole, the proteins were eluted with a linear gradient from 20 to 250 mM imidazole over 150 mL. Protein-containing fractions were pooled further purified by size exclusion chromatography using a self-packed Superose 6 16/60 column (Xn-XptA1, Yp-TcaATcaB) or a Sephacryl S400 column (GE Healthcare Life Sciences, Pl-TcdA1, Mm-TcdA4). Purified proteins were either directly used or supplemented with 35% glycerol and stored at −80 °C.

The expression plasmid for PNGase F, pOPH6, was a gift from Shaun Lott (Addgene plasmid #40315) and PNGase F was expressed and purified according to the supplemented protocol[44]. *E. coli* BL21DE3 was transformed with the expression plasmid and grown in 2 L LB medium at 37°. At an OD$_{600}$ of 0.5, 1 mM IPTG was added and the expression was carried out for 5 h at 37 °C. Cells were harvested and pelleted again in 100 mL of 0.1 M Tris (pH 8.0), 1 mM EDTA, 0.5 M sucrose (lysis buffer) at 3500 × *g*. The cell pellets were resuspended in 100 mL pure water, stirred for 10 min at 4 °C and subsequently 1 mM MgCl₂ was added. After an additional 10 min at 4 °C, the cells were pelleted again and the supernatant with the periplasmic proteins was supplemented with a final concentration of 20 mM MOPS-NaOH (pH 7.0). After 1.5 days of dialysis against 20 mM MOPS-NaOH (pH 8.0), 20 mM imidazole, 500 mM NaCl, PNGase F was purified using a 5 mL HisTrap FF column (GE Healthcare Life Sciences). The protein was eluted with a linear gradient from 20 to 400 mM imidazole over 60 mL. Protein-containing fractions were dialyzed against 50 mM potassium phosphate (pH 8.0) and used immediately afterwards.

**Labeling of TcAs for the glycan interaction screen**. Pl-TcdA1 was labeled with AlexaFluor488 C5 maleimide (Thermo Fisher, Cat. No. A10254) in a 1:3 molar ratio in 20 mM HEPES-NaOH (pH 7.3), 150 mM NaCl, 0.05% Tween-20 overnight at 4 °C. The other TcAs (Mm-TcdA4, Xn-XptA1, Yp-TcaATcaB) were labeled with AlexaFluor488 *N*-hydroxysuccinimide (NHS) ester (Thermo Fisher,

Cat. No. A20000) in a 1:3 molar ratio in 20 mM HEPES-NaOH (pH 7.3), 150 mM NaCl, 0.05% Tween-20 overnight at 4 °C. Subsequently, unreacted labeling dye was removed via size exclusion chromatography (SEC) on a Superose 6 Increase column (GE Healthcare Life Sciences) in 20 mM HEPES-NaOH (pH 7.3), 150 mM NaCl, 0.05% Tween-20. All labeled proteins were flash-frozen in liquid nitrogen in the presence of 35% glycerol and stored at −80 °C until usage.

**Glycan interaction microarray**. Amine-functionalized oligosaccharides or natural heparin (5 kDa) isolated from porcine intestinal mucosae (Sigma-Aldrich) were immobilized on commercial NHS ester-activated microarray slides (CodeLink Activated Slides; SurModics, Eden Prairie, MN, USA) using a piezoelectric spotting device (S3; Scienion, Berlin, Germany). Initial screening was performed with a comprehensive microarray displaying a large library of oligosaccharide antigens (Supplementary Table 2), as described before[45]. The HS/heparin microarray has been published before[46,47] (Supplementary Table 3). Other focused microarray slides that were used to confirm initial hits contained oligosaccharides related to Lewis X. Samples for spotting were diluted in 50 mM sodium phosphate buffer, pH 8.5. Spotted slides were incubated in a humid chamber for 24 h at room temperature to complete coupling reactions. Remaining NHS ester groups were deactivated with 50 mM ethanolamine in 50 mM sodium phosphate buffer, pH 9, for 1 h at 50 °C. Slides were rinsed three times with deionized water, dried by centrifugation (300 × *g*, 5 min) and stored desiccated until use. Spotted and quenched microarray slides were blocked using 100 mM HEPES-NaOH (pH 7.4), 200 mM NaCl, 1% (w/v) BSA (PBS-BSA) for 1 h at room temperature, washed three times with 100 mM HEPES-NaOH (pH 7.4), 200 mM NaCl, 0.05% (v/v) Tween-20 (wash buffer) and dried by centrifugation. FlexWell 16 or FlexWell 64 grids (Grace Bio-Labs, Bend, OR, USA) were applied to the slides to yield 16 or 64 wells for individual experiments, depending on the printing patterns. Slides were incubated with samples diluted in 100 mM HEPES-NaOH (pH 7.4), 200 mM NaCl, 0.01% (v/v) Tween-20 and 1% (w/v) BSA for 1 h at room temperature in a humid chamber. Wells were washed three times using wash buffer and rinsed once with deionized water. Grids were removed and slides were dried by centrifugation (300 × *g*, 5 min). The microarray slides were scanned with a GenePix 4300A scanner (Molecular Devices; Sunnyvale, CA, USA). The photomultiplier tube voltage was adjusted to reveal scans free of saturated signals. Image analysis was carried out with the GenePix Pro 7 software supplied with the instrument. Background-subtracted mean fluorescence intensity (MFI) values were exported for further analysis.

**Conjugation of Lewis X and BSA**. We prepared a glycoconjugate composed of Lewis X trisaccharide (compound 7, Fig. 2c) and BSA following modified procedures described before[48]. The procedure is illustrated in Supplementary Fig. 2b. First, **7** was combined with a 6-fold molar excess of di-*p*-nitrophenyl adipate in 400 μL anhydrous dimethyl sulfoxide (DMSO)/pyridine (2:1) and 10 μL triethylamine (Et₃N). The reaction mixture was incubated for 2 h at room temperature while stirring. Solvents were evaporated by lyophilization. Dried reaction products were successively washed with dichloromethane and chloroform (ten times 1 mL each) until thin-layer chromatography (TLC) revealed complete removal of non-reacted crosslinker. The washed reaction products were solubilized in DMSO, transferred to new reaction tubes and lyophilized. Dried products were reacted with bovine serum albumin (BSA; Pan Biotech) in 100 mM sodium phosphate buffer, pH 8, for 24 h at room temperature while stirring. The resulting glycoconjugate was washed and concentrated with deionized water using 10 kDa centrifugal filter units (Merck Millipore, Tullagreen, Ireland).

Conjugation was confirmed by MALDI–TOF MS using an Autoflex Speed system (Bruker Daltonics; Bremen, Germany). Samples were spotted using the dried droplet technique with 2,5-dihydroxyacetophenone (DHAP) as matrix on MTP 384 ground steel target plates (Bruker Daltonics). Samples were prepared by mixing 2 μL of desalted protein sample with 2 μL of DHAP matrix and 2 μL of 2% (v/v) trifluoroacetic acid (TFA) prior to spotting. The mass spectrometer was operated in linear positive mode. Mass spectra were acquired over an $m/z$ range from 30,000 to 210,000 and data was analyzed with the FlexAnalysis software provided with the instrument. The mass shift of about 14,500 Da showed that, on average, 20 molecules of Lewis X were conjugated per molecule of BSA (Supplementary Fig. 2c).

**Biolayer interferometry (BLI).** The affinity of Pl-TcdA1 to BSA-Lewis X was determined by BLI using an Octet RED384 (forteBio, Pall Life Sciences) and streptavidin biosensors. BSA-Lewis X was biotinylated in 20 mM HEPES-NaOH (pH 7.3), 150 mM NaCl, 0.05% Tween 20 (labeling buffer) with NHS-LC-Biotin (Thermo Fisher, Cat. No. 21336) in a 1:3 molar ratio for 2 h at room temperature. Unreacted biotin label was removed via SEC on a Superdex 200 Increase column (GE Healthcare Life Sciences). Biotinylated BSA-Lewis X was immobilized on streptavidin biosensors at 10 μg mL$^{-1}$, and Pl-TcdA1 pentamer concentration in solution was 25–1600 nM. BLI sensorgrams were measured in three steps: baseline (300 s), association (90 s), and dissociation (60 s). The sensorgrams were corrected for background association of Pl-TcdA1 on unloaded streptavidin biosensors. On- and off-rates of TcA binding were determined simultaneously by a global curve fit according to a 1:1 binding model. All BLI steps were performed in labeling buffer with 0.3 mg mL$^{-1}$ BSA.

The affinity of Mm-TcdA4 to heparin was determined analogously using the same instrument setup and buffers as described above. Biotinylated porcine intestinal mucosa heparin (Merck Millipore) was immobilized on streptavidin biosensors at 10 μg mL$^{-1}$, and Mm-TcdA4 pentamer concentration in solution was 8–500 nM. BLI sensorgrams were measured with 600 s association and 600 s dissociation time, and corrected for background association of Mm-TcdA4 on unloaded streptavidin biosensors. On- and off-rates of TcA binding were determined simultaneously by a global curve fit according to a 1:1 binding model.

**Deglycosylation of HEK 293T cells.** $5.5 \times 10^6$ HEK 293T cells (Thermo Fisher) attached to 10 cm cell culture dishes (Sarstedt) in DMEM/F12 + 10% FBS medium were incubated with 20 μg PNGase F (self-prepared for flow cytometry; NEB Cat. No. P0704 for confocal microscopy) for 5 h at 37 °C. Subsequently, cells were detached from the surface by carefully resuspending them, washed once in DMEM/F12 + FBS and resuspended in 1.8 mL PBS immediately before subsequent intoxication and flow cytometry.

**Flow cytometry.** Immediately after deglycosylation and washing, HEK 293T or HEK 293 GnTI$^-$ cells (Thermo Fisher) were incubated with 200 nM (monomer concentration) Pl-TcdA1 labeled with AlexaFluor488 for 15 min. Subsequently, cells were washed one time, resuspended in PBS without toxin and injected into a LSRII flow cytometer (BD Biosciences). 50000 cells were counted in total. Cell fluorescence caused by Pl-TcdA1 binding was analyzed using FlowJo.

**Intoxication assay.** HEK 293T cells or HEK 293 GnTI$^-$ cells were intoxicated with pre-formed holotoxin formed by Pl-TcdA1 and Pl-TcdB2–TccC3. Cells ($5 \times 10^4$) were grown adherently in 400 μL DMEM/F12 medium (Pan Biotech) overnight and 0.5 or 2 nM holotoxin was subsequently added. Incubation was performed for 16 h at 37 °C before imaging. Experiments were performed in triplicate. Cells were not tested for *Mycoplasma* contamination.

**Confocal microscopy.** Prior to confocal microscopy experiments, cells (HEK 293T and HEK 293 GnTI$^-$) were seeded at a density of $0.3 \times 10^6$ cells per dish and grown on 35 mm poly-D-lysine coated glass-bottom culture dishes (MatTek) until 30–40% confluency in 2 mL DMEM/F12 + 10% FBS for 36 h in a 37 °C, 5% CO$_2$ humidified atmosphere. Confocal imaging was carried out at 37 °C in a 5% CO$_2$ atmosphere using a ZEISS LSM 800 microscope equipped with a C-Apochromat 40×/1.2 W objective. After the initial image was taken (0 h), AlexaFluor488 labeled Pl-TcdA1 was added to a final concentration of 500 pM. Images were then taken at 10-min intervals for 8 h. The experiments were done in three biological replicates for each cell type.

Images taken at 1 h intervals were processed in Fiji[49]. A cell mask was generated by using the Trainable Weka Segmentation plugin[50] on the transmitted light channel and then used to focus on the relevant areas of the fluorescent channel. The mean gray values from the fluorescent channel were then measured for each image, which represent the sum of all gray pixel values (fluorescence) in the selections divided by the total number of pixels (cell surface). The autofluorescence for each replicate was removed by subtracting the mean gray value of the initial 0 h image from later post-intoxication images. The averages and standard deviation for this normalized mean fluorescence intensity were then calculated and plotted.

**Crosslinking of Pl-TcdA1 and BSA-Lewis X.** Pl-TcdA1 (1 μM) and BSA-Lewis X (30 μM) were incubated in 20 mM HEPES-NaOH (pH 8.0), 150 mM NaCl, 0.02% Tween-20 overnight at 4 °C. Subsequently, glutaraldehyde was added to a final concentration of 0.1% and proteins were crosslinked for 90 min at 22 °C. The reaction was stopped with 1.7 mM Tris–HCl (pH 8.0) and crosslinked Pl-TcdA1-BSA-Lewis X was purified from excess of BSA-Lewis X by SEC using a Superose 6 3.2-300 column (GE Healthcare Life Sciences).

**Preparation and data acquisition of Pl-TcdA1-BSA-Lewis X.** 4 μL of 0.06 mg mL$^{-1}$ crosslinked complex were applied to glow-discharged holey carbon grids (Quantifoil R1.2/1.3, 300 mesh) covered with a 2 nm carbon layer. After 10 min of incubation time on the grid, the grid was manually blotted and another 4 μL were applied and incubated for 20 s. Subsequently, the sample was vitrified in liquid ethane with a Cryoplunge3 (Cp3, Gatan) using 1.8 s blotting time at 90% humidity.

A dataset of Pl-TcdA1–BSA-Lewis X was collected using a Cs corrected Titan Krios equipped with an XFEG and a Falcon III direct electron detector. Images were recorded using the automated acquisition program EPU (FEI) at a magnification of 59,000×, corresponding to a pixel size of 1.11 Å per pixel on the specimen level. 4992 movie-mode images were acquired in a defocus range of 1.0–2.5 μm. Each movie comprised 40 frames acquired over 1.5 s with a total cumulative dose of ~100e$^-$ per Å$^2$.

**Image processing of Pl-TcdA1-BSA-Lewis X.** After initial screening of all micrographs, 3878 integrated images were selected for further processing. Movie frames were aligned, dose-corrected, and averaged using MotionCor2 (ref. [51]). CTF parameters were estimated using CTER[52], implemented in the SPHIRE software package[30]. Initially, 4300 particles were manually picked and 2D class averages used as an autopicking template were generated using ISAC[53] in SPHIRE. 711,872 particles were auto-picked from the images using Gautomatch[54] and extracted from the micrographs.

Reference-free 2D classification and cleaning of the dataset was performed with the iterative stable alignment and clustering approach ISAC in SPHIRE. ISAC was performed with a pixel size of 5.61 Å per pixel on the particle level. The 'Beautifier' tool of SPHIRE was then applied to obtain refined and sharpened 2D class averages at the original pixel size, showing toxin particles with visible additional density at the TcA pentamer, which corresponds to BSA-Lewis X (Fig. 3a). 199,038 particles were selected for subsequent 3D refinement. We applied our previous reconstruction of Pl-TcdA1 (EMD-3645) as initial model after scaling and filtering it to 25 Å resolution and performed an initial 3D refinement in SPHIRE using Meridien with C5 symmetry. The density corresponding to the bound BSA-Lewis X appears only at very low binarization threshold (Supplementary Fig. 4). Therefore, we expanded the symmetry of the input particle to C5 via the "symmetrize" option of SPHIRE, resulting in 995,190 particles. Subsequently, we applied 3D sorting (Sort3D in SPHIRE) to the symmetrized stack using a focused mask that comprises the lower part of the shell and the additional densities. The resulting 3D classes showed 1–2 additional densities corresponding to BSA-Lewis X at different orientations. We next rotated the projection parameters of the classes by +72° or −72° in order to orient the most pronounced additional density to the same direction in all classes (Supplementary Fig. 4). Finally, after import of the rotated projection parameters and stack de-symmetrization, we ran a local refinement in Meridien with C1 symmetry. The resolution of the final density was estimated to 7.0/5.0 Å according to FSC 0.5/0.143 after applying a soft Gaussian mask. The B-factor was set to −80.0 Å$^2$ for sharpening.

**Sample preparation and data acquisition of Mm-TcdA4/heparin.** Incubation of Mm-TcdA4 with porcine mucosa heparin in solution resulted in precipitation. Therefore, 3 μL of 0.10 mg mL$^{-1}$ Mm-TcdA4 in 20 mM Tris–HCl (pH 8.0), 250 mM NaCl, 0.05% Tween-20 was pre-applied on a glow-discharged holey carbon grid (Quantifoil, R2/1, 300 mesh) covered with a 2 nm carbon layer. After 20 s of incubation time on the grid, the grid was manually blotted and 3 μL of 0.3 mg mL$^{-1}$ heparin from porcine intestinal mucosa (Millipore, Cat. No. 375095) applied and incubated for 2 min at 22 °C. Subsequently, the sample was vitrified in liquid ethane with a Cryoplunge3 (Cp3, Gatan) using 1.8 s blotting time at 89% humidity.

A dataset of Mm-TcdA4/Heparin was collected at the Max Planck Institute of Molecular Physiology, Dortmund using a Cs corrected Titan Krios equipped with an XFEG and a Falcon III direct electron detector. Images were recorded using the automated acquisition program EPU (FEI) at a magnification of 59,000×, corresponding to a pixel size of 1.11 Å per pixel on the specimen level. 5549 movie-mode images were acquired in a defocus range of 1.2–2.2 μm. Each movie comprised 40 frames acquired over 1.5 s with a total cumulative dose of ~100e$^-$ per Å$^2$.

**Image processing of Mm-TcdA4/heparin.** After initial screening of all micrographs, 4749 integrated images were selected for further processing. Movie frames were aligned, dose-corrected, and averaged using MotionCor2 (ref. [51]). The integrated images were also used to determine the contrast transfer function (CTF) parameters with CTER[52], implemented in the SPHIRE software package[30]. Initially, 4300 particles were manually picked and 2D class averages used as an autopicking template were generated using ISAC[53] in SPHIRE. 477,602 particles

were auto-picked from the images using Gautomatch[54] and extracted from the micrographs.

Reference-free 2D classification and cleaning of the dataset were performed with the iterative stable alignment and clustering approach ISAC in SPHIRE. ISAC was executed with a pixel size of 5.61 Å per pixel on the particle level. The 'Beautifier' tool of SPHIRE was then applied to obtain refined and sharpened 2D class averages at the original pixel size, showing high-resolution features (Supplementary Fig. 6b). From the initial set of particles, the clean set used for 3D refinement contained 182,506 particles. We applied our previous 3.3 Å reconstruction of Mm-TcdA4 (ref. [23]) (EMD-10035) as an initial model after scaling and filtering it to 25 Å resolution and performed 3D refinement in SPHIRE using Meridien with imposed C5 symmetry. The resolution of the final density was estimated to be 3.6/3.2 Å according to FSC 0.5/0.143 after applying a soft Gaussian mask. The B-factor was estimated to be 127.6 Å². Local FSC calculation was performed using the Local Resolution tool in SPHIRE. (Supplementary Fig. 6f) and the density map was filtered according to its local resolution using the 3-D Local Filter tool in SPHIRE.

The local resolution of the additional density corresponding to heparin was however not sufficient to obtain a molecular model. We therefore filtered the map obtained here and the Mm-TcdA4 map without heparin (EMDB 10035) to 5 Å resolution, normalized the maps and subtracted the filtered and normalized Mm-TcdA4 map without heparin from the map with heparin. The difference density was filtered to 22 Å resolution to visualize the density corresponding to heparin (Supplementary Fig. 6g).

**Sample preparation and data acquisition of Xn-XptA1/heparin**. Analogous to Mm-TcdA4, 0.1 mg mL⁻¹ Xn-XptA1 in 20 mM Tris–HCl (pH 7.5), 150 mM NaCl, 0.05% Tween-20 were first applied onto a glow-discharged grid (Quantifoil gold R2/1, 300 mesh) covered with a 2 nm carbon layer. After 30 s of incubation time on the grid, the grid was manually blotted and 4 μL of 0.3 mg mL⁻¹ heparin from porcine intestinal mucosa (Millipore, Cat. No. 375095) were applied and incubated for 4 min at 12 °C. Subsequently, the sample was vitrified in liquid ethane with a Vitrobot Mark IV (Thermo Fisher) using 3.0 s blotting time at 100% humidity.

A dataset of Xn-XptA1/heparin was collected at the Max Planck Institute of Molecular Physiology, Dortmund using a Talos Arctica transmission electron microscope equipped with an XFEG and a Falcon III direct electron detector. Images were recorded using the automated acquisition program EPU (FEI) at a magnification of 120,000×, corresponding to a pixel size of 1.21 Å per pixel on the specimen level. 1855 movie-mode images were acquired in a defocus range of 1.5–2.5 μm. Each movie comprised 40 frames acquired over 4.0 s with a total cumulative dose of ∼ 52e⁻ per Å².

**Image processing of Xn-XptA1/heparin**. Movie frames were aligned, dose-corrected, and averaged using MotionCor2 (ref. [51]). The integrated images were also used to determine the CTF parameters with CTFFIND4 (ref. [55]). 346,048 particles were auto-picked from the images using crYOLO with a general picking model[56], and 315,930 particles were extracted from the micrographs.

Reference-free 2D classification and cleaning of the dataset were performed with the iterative stable alignment and clustering approach ISAC in SPHIRE. ISAC was executed with a pixel size of 6.37 Å per pixel on the particle level. The 'Beautifier' tool of SPHIRE was then applied to obtain refined and sharpened 2D class averages at the original pixel size, showing high-resolution features (Supplementary Fig. 7b). From the initial set of particles, the clean set used for 3D refinement contained 172,596 particles. We applied our previous 2.8 Å reconstruction of Xn-XptA1 (ref. [23]) (EMD-10034) as an initial model after scaling and filtering it to 20 Å resolution and performed 3D refinement in SPHIRE using Meridien with imposed C5 symmetry. The resolution of the final density was estimated to be 4.3/3.7 Å according to FSC 0.5/0.143 after applying a soft Gaussian mask. The B-factor was estimated to be 65.8 Å². Local FSC calculation was performed using the Local Resolution tool in SPHIRE. (Supplementary Fig. 7f) and the density map was filtered according to its local resolution using the 3-D Local Filter tool in SPHIRE.

Analogous to Mm-TcdA4, we filtered the maps of Xn-XptA1 (EMDB 10034) and Xn-XptA1/heparin to 5 Å resolution, normalized the maps and subtracted the filtered and normalized Xn-XptA1 map without heparin from the map with heparin. The difference density was filtered to 22 Å resolution to visualize the density corresponding to heparin/HS (Supplementary Fig. 7g).

**Atomic model building and refinement**. For Pl-TcdA1, we rigid-body fitted the model (PDB 6RW6) into the 5.0 Å density map of Pl-TcdA1-BSA-Lewis X in UCSF Chimera[57]. For Mm-TcdA4 and Xn-XptA1, we first rigid-body fitted the models (PDB 6RW9 and 6RW8, respectively) into the density maps of Mm-TcdA4/heparin and Xn-XptA1/heparin using UCSF Chimera and subsequently refined the models with the real space refinement tool of PHENIX[58] and Rosetta relaxation[59]. The geometries of the refined models were validated using MolProbity, and the data statistics are summarized in Supplementary Table 1.

**Molecular docking**. We used AutoDock Vina[60] via the UCSF Chimera plugin to visualize potential conformations of heparin bound to Mm-TcdA4 and Xn-XptA1,

respectively, and Lewis X bound to Pl-TcdA1. Before docking, the interface regions of the proteins were protonated using H++[61] at pH 7.0 and 150 mM ionic strength. For Mm-TcdA4 and Xn-XptA1, we docked a heparin pentasaccharide (molecule code NTP) after geometry optimization in phenix.elbow[62] to the α-helical shell region of Mm-TcdA4 or to the gap between the neuraminidase-like domain, RBD B and RBD D in Xn-XptA1. We chose the conformations from the obtained docking results that fitted best into the respective additional density. The resulting scores of the docking poses were −6.2 kcal mol⁻¹ for Mm-TcdA4 and −6.1 kcal mol⁻¹ for Xn-XptA1, respectively.

For Pl-TcdA1, we docked the Lewis X trisaccharide to RBD D (residues 1638–1754). We chose two representative conformations which are closely located to the additional density with scores of −4.5 kcal mol⁻¹ each.

**Reporting summary**. Further information on experimental design is available in the Nature Research Reporting Summary linked to this paper.

## Data availability
The coordinates for the EM structures of Pl-TcdA1/BSA-Lewis X, Mm-TcdA4/heparin, and Xn-XptA1/heparin have been deposited in the Electron Microscopy Data Bank under accession numbers 10794, 10796, and 10797, respectively. The models of Mm-TcdA4 refined against the EM density map of Mm-TcdA4/heparin and Xn-XptA1 refined against the EM density map of Xn-XptA1/heparin have been deposited in the Protein Data Bank under accession numbers 6YEW and 6YEY, respectively. The source data underlying Figs. 1b, c, 2a, b, and 4a and Supplementary Figs. 2d and 5a, b are provided as a Source Data file. Other data are available from the corresponding author upon reasonable request.

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

## Acknowledgements

The authors thank Dr. O. Hofnagel and Dr. D. Prumbaum for excellent assistance in electron microscopy and acquisition of the datasets. The authors thank A. Roderer and K. Vogel-Bachmayr for purification of Pl-TcdA1, Mm-TcdA4, and Pl-TcdB2-TccC3, respectively, and M. Hülseweh for preparation of HEK cells. The authors would like to thank Dr. T. Wagner for aiding with Fiji, and M. Schulz for helping with flow cytometry. This work was supported by funds from the Max Planck Society (to S.R. and P.H.S.) and the European Research Council under the European Union's Seventh Framework Programme (FP7/2007-2013) (Grant No. 615984, to S.R.).

## Author contributions

S.R. and P.H.S. designed the project. D.R., O.S., and F.L. prepared fluorescently labeled TcAs for the glycan array and cellular experiments. F.B. and P.K. carried out the glycan arrays and interpreted the data. F.B. prepared the BSA-Lewis X adduct. D.R. analyzed the affinities of proteins to glycans, performed flow cytometry experiments, prepared cryo-EM samples, processed and analyzed the cryo-EM data. O.S. recorded and analyzed confocal microscopy images. D.R., O.S., F.B., and P.K. prepared the figures. All authors discussed the results and contributed in writing of the manuscript.

## Competing interests

The authors declare no competing interests.
