## [Peer Review File · Nature Communications]

Reviewers' comments:

Reviewer #1 (Remarks to the Author):

Tc toxins are a type of toxin found in insect and human bacterial pathogens. Tc toxins are composed of a pentameric membrane translocation component (TcA) and a cocoon-shaped TcB and TcC. The toxic enzyme that is held in the cocoon enters a translocation channel created by TcA. Once the holotoxin is bound to cell membrane it is endocytosed and acidification in the late endosome triggers the opening of the cocoon shell and the formation of a membrane channel that allows for the translocation of the toxin into the host cell. There have been structures of the holotoxin in various pre-pore and pore conformations (most determined by this laboratory); however, less is understood how the toxin is recruited and engaged at the cellular membrane. This manuscript focuses on isolating Tc holotoxin cell surface receptors and then understanding how the Tc toxin binds to the receptors. The authors identify heparins and Lewis antigens as receptors for Tc toxins from insect- and human pathogenic bacteria. They then use glycan arrays and cryo-EM structural analyses to characterize how the identified receptors interact with the toxins. These analyses shows that the glycans (heparins/heparan sulfates) bind TcAs with the alpha-helical part of the shell domain, while the Lewis antigen (predicted to be similar to glycosylation found on receptors) interacts with the receptor-binding domain of the toxin of *P. luminescens* TcdA1. From these results the authors propose a two-step association model, at least for PI TcdA1, where first the toxin interact with the heparin glycans then is able to bind the Lewis antigen (Lewis X). This paper presents strong experiments and structures that provide new mechanistic information on how Tc toxins interact with cellular membranes.

Suggestions:

1. All the Tc toxins tested in this study appear to bind heparins, but only PI TcdA1 appears to bind the Lewis-X antigen. There are a number of Tc toxin structures that have been determined and definitely the sequences are known. The authors should comment in the discussion on whether the binding pockets they have identified in this study exist (or are predicted to exist) in other Tc toxins (i.e. will this be a conserved mechanism for all Tc toxins or only a subset). This would provide a broader context to their proposed "two-step" model for Tc toxin interaction with the membrane.
2. The authors use mammalian cell lines to test for binding to glycans. Is there a reason why the experiments were not done with bacteria instead – since these toxins infect bacteria?
3. In Figure 1, the cell microscopy experiments are done at 4h and 8h time points. This would measure endocytosis (and intracellular degradation?) rather than the ability to interact with cell membrane interaction. From the methods it appears the FACs sorting is done only 15 minutes. Were the cells kept on ice to inhibit endocytosis? The authors should comment on how they are sure that their cellular assays (especially at the 4 and 8 hour time point) are accurately measuring the ability of the Tc toxin to interact with membrane, rather than being endocytosed and degraded within the cell.

Reviewer #2 (Remarks to the Author):

This paper sets out to determine the receptors responsible for cell recognition by ABC-type Toxin Complexes (Tc). Using glycan arrays, the authors determine that both Lewis antigens and sulfated heparans can bind to different sites on the A subunit of the Tc, and therefore propose a two-step mechanism, whereby the TC first binds to heparan sulfate and then to a Lewis antigen sugar. This subject is an important one, as the cell-surface receptors for the ABC Tcs are unknown, and their identification remains a key bottleneck in understanding the cell tropism of this family of bacterial toxins.

The first half of the paper describes the identification of a subset of Lewis antigen oligosaccharides as specific ligands for the Receptor Binding Domain (RBD) D of the TcdA subunit of the Tc from *Photobacterium luminescens*. The work is both carefully carried out and convincing, and it is clear that the core unit recognised is the Lewis X trisaccharide, and that the beta-4 linkage between N-Acetylglucosamine and galactose is essential for tight binding.

A few things would improve this section:

1) It should be made clearer exactly which saccharide is immobilised to BSA for binding experiments - the Lewis X trisaccharide (compound 7). This detail is buried in the methods section and should be more clearly stated in the results and in Figure 2.

2) p7, line 140 - high amounts or high concentrations? I suspect the latter is what is meant.

3) The binding site for the trisaccharide is reasonably well described in Figure 3d and in the text, but I feel a great deal more information could be readily obtained by some more comparative analysis. For example, is the identified binding site also seen in the other RBDs of the P. luminescens A protein? Is the site conserved in the A subunits from other species? Lewis X sugars are known to be receptor binding sites for other bacterial pathogens. How does the identified binding site compare to other known examples of Lewis X binding?

The second half of the paper describes the identification of heparin and heparan sulfate binding to several TcA subunits from different species. In comparison with the first half of the paper, this second section has a number of problems. To begin with, the rationale for choosing to screen heparan oligosaccharides is not clearly described. The initial broad screen is a self-evident place to start, but the rationale for focussing on heparans is not clear, especially as the initial screen contained heparans (compounds 183-186), but they did not show any binding. In fact, the binding of the P. luminescens TcA in Figure 4 seems at odds with its lack of binding to the same sugars in Figure 2, though it is difficult to be clear about this as the numbering scheme for the compounds does not seem to be consistent.

The binding of TcA subunits is strongly electrostatic, and I wonder if enough controls have been put in place to be sure that it is specific? What are the pI values for BSA and WGA? Do either of them contain surface patches containing K or R residues? The inclusion of a known heparan binding protein as a positive control would help put the level of binding signal in context.

In contrast to the data presented for the Lewis X binding, the quantitation of heparin binding by BLI is not reliable. The fitting of the model 1:1 binding model to the data in Supplementary Figure 5 has failed, and the sub-nM estimate of KD is almost certainly wrong. More than that, the dose-response curve for binding is not credible, with 63, 125 and 250 nM concentrations of heparin all giving the same equilibrium binding response. Would it be possible to quantitate binding (using BLI or SPR) with a defined heparan sulfate sugar? This might remove some of the uncertainty introduced by using heparin.

The identified heparin binding site is a surprise, given that it is in the alpha-helical domain of the shell of TcA, rather than in any of the putative RBDs. Is the positively charged cleft identified in Mm-TcdA4 conserved in the other TcA subunits? Is that area also seen in Type II (chitin containing) Tcs?

Given these reservations about the heparin binding, I am somewhat skeptical about the proposed two-step binding model. In particular, the idea that heparin binding is the initial contact with a sub-nM KD seems at odds with the idea that the uM binding of the Lewis antigens allows laterally mobile, dynamic binding of TcA to the cell surface.

Finally, it is not made clear if the observed glycan binding patterns are congruent with, and make

a rational explanation for, the known cell tropism of different ABC toxins for different eukaryotic cell types.

Shaun Lott, University of Auckland.

We thank the reviewers for their constructive and positive feedback, which aided us to further improve the manuscript. Major modifications of the manuscript are highlighted in yellow. Below we include our detailed response to each point.

Reviewers' comments:

Reviewer #1 (Remarks to the Author):

Tc toxins are a type of toxin found in insect and human bacterial pathogens. Tc toxins are composed of a pentameric membrane translocation component (TcA) and a cocoon-shaped TcB and TcC. The toxic enzyme that is held in the cocoon enters a translocation channel created by TcA. Once the holotoxin is bound to cell membrane it is endocytosed and acidification in the late endosome triggers the opening of the cocoon shell and the formation of a membrane channel that allows for the translocation of the toxin into the host cell. There have been structures of the holotoxin in various pre-pore and pore conformations (most determined by this laboratory); however, less is understood how the toxin is recruited and engaged at the cellular membrane. This manuscript focuses on isolating Tc holotoxin cell surface receptors and then understanding how the Tc toxin binds to the receptors. The authors identify heparins and Lewis antigens as receptors for Tc toxins from insect- and human pathogenic

bacteria. They then use glycan arrays and cryo-EM structural analyses to characterize how the identified receptors interact with the toxins. These analyses shows that the glycans (heparins/heparan sulfates) bind TcAs with the alpha-helical part of the shell domain, while the Lexis antigen (predicted to be similar to glycosylation found on receptors) interacts with the receptor-binding domain of the toxin of *P. luminescens* TcdA1. From these results the authors propose a two-step association model, at least for PI TcdA1, were first the toxin interact with the heparin glycans then is able to bind the Lewis antigen (Lewis X). This is paper presents strong experiments and structures that provide new mechanistic information on how Tc toxins interact with cellular membranes.

Suggestions:

1. All the Tc toxins tested in this study appear to bind heparins, but only PI TcdA1 appears to bind the Lexis-X antigen. There are a number of Tc toxin structures that have been

determined and definitely the sequences are known. The authors should comment in the discussion on whether the binding pockets they have identified in this study exist (or are predicted to exist) in other Tc toxins (i.e. will this be a conserved mechanism for all Tc toxins or only a subset). This would provide a broader context to their proposed “two-step” model for Tc toxin interaction with the membrane.

We thank reviewer 1 for this excellent suggestion and added an additional Supplementary Figure (Supplementary Fig. 8) that shows a structure-based sequence alignment of RBD-D and the docking of the Lewis X trisaccharides into the respective binding pockets. Most residues of RBD D including those forming the potential binding pockets are not conserved. We added the detailed paragraph to the discussion (1.255ff).

2. The authors use mammalian cell lines to test for binding to glycans. Is there a reason why the experiments were not done with bacteria instead – since these toxins infect bacteria?

Tc toxins do not attack bacteria, but are rather produced by insect pathogens such as *P. luminescens* or by human pathogens such as *M. morgani*. Therefore, mammalian cells like the HEK293 cells that were used here are suitable systems, and have been shown to be highly sensitive to Tc toxins²⁻⁴. Since the two described toxic enzymes of Tc target the eukaryotic proteins actin and Rho², using bacterial targets for the binding/intoxication assays would not make sense.

3. In Figure 1, the cell microscopy experiments are done at 4h and 8h time points. This would measure endocytosis (and intracellular degradation?) rather than the ability to interact with cell membrane interaction. From the methods it appears the FACs sorting is done only 15 minutes. Were the cells kept on ice to inhibit endocytosis? The authors should comment on how they are sure that their cellular assays (especially at the 4 and 8 hour time point) are accurately measuring the ability of the Tc toxin to interact with membrane, rather than being endocytosed and degraded within the cell.

The images in Fig. 1a were taken after incubation at 37 °C, as stated in the Methods section. We have added this information to the figure legend.

Although the cells endocytosed a large fraction of the toxin after 4 h and 8 h (intensely fluorescent intracellular bodies), there is still fluorescence visible on the cell surface. This indicates continuous binding and accumulation of Tc over the entire incubation period. The cells are not killed and continue taking up toxin because PI-TcdA1 alone is not toxic at 500 pM, which enhances the overall fluorescence with increasing incubation times. Since cell membrane binding has to precede endocytosis and since we quantified the overall fluorescence associated to cells (Fig. 1b), it does not matter for our measurements if the toxin is still on the cell surface or inside the cells after endocytosis. The same argument applies to the flow cytometry experiments.

Reviewer #2 (Remarks to the Author):

This paper sets out to determine the receptors responsible for cell recognition by ABC-type Toxin Complexes (Tc). Using glycan arrays, the authors determine that both Lewis antigens and sulfated heparans can bind to different sites on the A subunit of the Tc, and therefore propose a two-step mechanism, whereby the TC first binds to heparan sulfate and then to a Lewis antigen sugar. This subject is an important one, as the cell-surface receptors for the ABC Tcs are unknown, and their identification remains a key bottleneck in understanding the cell tropism of this family of bacterial toxins.

The first half of the paper describes the identification of a subset of Lewis antigen oligosaccharides as specific ligands for the Receptor Binding Domain (RBD) D of the TcdA subunit of the Tc from *Photobacterium luminescens*. The work is both carefully carried out and convincing, and it is clear that the core unit recognised is the Lewis X trisaccharide, and that the beta-4 linkage between N-Acetylglucosamine and galactose is essential for tight binding.

A few things would improve this section:

1) It should be made clearer exactly which saccharide is immobilised to BSA for binding experiments - the Lewis X trisaccharide (compound 7). This detail is buried in the methods section and should be more clearly stated in the results and in Figure 2.

We thank reviewer 2 for pointing this out. We have added the information to the legends of Figure 2, Supplementary Figure 2 and to the results part (1.124).

2) p7, line 140 - high amounts or high concentrations? I suspect the latter is what is meant.

We thank reviewer 2 for pointing out this issue and corrected our phrasing to “high concentrations”.

3) The binding site for the trisaccharide is reasonably well described in Figure 3d and in the text, but I feel a great deal more information could be readily obtained by some more comparative analysis. For example, is the identified binding site also seen in the other RBDs of the *P. luminescens* A protein? Is the site conserved in the A subunits from other species? Lewis X sugars are known to be receptor binding sites for other bacterial pathogens. How does the identified binding site compare to other known examples of Lewis X binding?

We thank this reviewer for this excellent suggestion. Please see the comment to point 1 of reviewer 1.

The second half of the paper describes the identification of heparin and heparan sulfate binding to several TcA subunits from different species. In comparison with the first half of the paper, this second section has a number of problems. To begin with, the rationale for choosing to screen heparan oligosaccharides is not clearly described. The initial broad screen is a self-evident place to start, but the rationale for focussing on heparans is not clear, especially as the initial screen contained heparans (compounds 183-186), but they did not show any binding. In fact, the binding of the *P. luminescens* TcA in Figure 4 seems at odds with its lack of binding to the same sugars in Figure 2, though it is difficult to be clear about this as the numbering scheme for the compounds does not seem to be consistent.

We thank reviewer 2 for pointing out that we did not describe our rationale why we chose the heparin screen to identify more Tc toxin interaction partners. We have now added introductory sentences to the results part (l.167ff and l.180ff).

Indeed, PI-TcdA1 shows only a weak binding signal for the glycan IdoA-2,4-disulfate(α -1)aminopentanol (compound 183) in the glycan screen in Fig. 2a. The binding intensity is below the arbitrary threshold of MFI=500. Also in the heparin screen (Fig. 4a and Supplementary Fig.5), PI-TcdA1 shows only a weak binding signal to this compound of an MFI below 500, in contrast to the stronger binding signals of Mm-TcdA4 and Xn-XptA1. PI-TcdA1 shows only pronounced binding to heparins 1, 2, and 11, here the binding signals are at least 3 times higher than the one for compound 183. Therefore, the results of the binding assays for PI-TcdA1 are consistent.

The glycan IDs including compounds 183 – 186 in Supplementary Tables 2 and 3 (first column) are consistent, and the respective positions on the heparin array in Supplementary Figure 5 are indicated (fourth column).

We address the differences in heparin binding intensities for the individual TcAs in 1.185ff (please see below).

The binding of TcA subunits is strongly electrostatic, and I wonder if enough controls have been put in place to be sure that it is specific? What are the pI values for BSA and WGA? Do either of them contain surface patches containing K or R residues? The inclusion of a known heparan binding protein as a positive control would help put the level of binding signal in context.

The binding signals that were obtained for Mm-TcdA4 and Xn-XptA1 were in the same range as those obtained for adenovirus-2, which had been screened on the same array layout⁷. The binding signals of Pl-TcdA1 and Yp-TcaATcaB were weaker and reached up to 40% of the values of Mm-TcdA4 for identical heparins. We added this information to the results section (1.185ff).

The control proteins were chosen to represent one basic and one acidic protein with pI values of 5.6 for BSA and 7.7 for WGA, respectively. Although the pI of WGA is higher than that of Mm-TcdA4 (pI of 5.8), WGA does not have pronounced positively charged patches since the K and R residues are evenly distributed over its surface. Also BSA does not have pronounced positively charged surface patches (see below).

Surface representations of Mm-TcdA4, WGA and BSA. The density according to heparin is shown in semi-transparent representation. The surfaces are colored according to the Coulomb potential with blue representing positive and red representing negative charges, analogous to Fig. 4d in the manuscript.

In contrast to the data presented for the Lewis X binding, the quantitation of heparin binding by BLI is not reliable. The fitting of the model 1:1 binding model to the data in Supplementary Figure 5 has failed, and the sub-nM estimate of K_D is almost certainly wrong. More than that, the dose-response curve for binding is not credible, with 63, 125 and 250 nM concentrations of heparin all giving the same equilibrium binding response. Would it be possible to quantitate binding (using BLI or SPR) with a defined heparan sulfate sugar? This might remove some of the uncertainty introduced by using heparin.

We thank this reviewer for pointing this out. We agree that the fit, representing a 1:1 binding model, is not reliable and therefore have removed it and the $K(D)$ value from the figure. Although a 1:1 binding model does not describe the reaction, the signal increase upon association and signal decrease upon dissociation prove that the ligand (Mm-TcdA4) interacts with the immobilized analyte (heparin). We rewrote the corresponding section of the figure legend. We also rewrote the corresponding part of the results section (1.202ff).

Since the defined heparins can only be produced in small amounts for the glycan array, we do not have access to them in quantities sufficient for biotinylation and biosensor immobilization.

The identified heparin binding site is a surprise, given that it is the alpha-helical domain of the shell of TcA, rather than in any of the putative RBDs. Is the positively charged cleft identified in Mm-TcdA4 conserved in the other TcA subunits? Is that area also seen in Type II (chitin containing) Tcs?

The positively charged cleft at the α -helical shell of Mm-TcdA4 is not conserved in other chitinase-independent TcAs and also not in the chitinase-dependent Yen-TcA⁸, which is in accordance with a general low conservation of the surface of TcA in comparison to the linker and the channel⁹. To assess whether heparin binding occurs at the same or different site, we solved the structure of another TcA, Xn-XptA1, together with heparin (Supplementary Fig.7). Here, we identified additional density not at the same site of the α -helical shell domain, but instead in a gap between the neuraminidase-like domain, RBD B and RBD D (Fig. 4f-h). This shows that, although heparin binding appears to occur in several TcAs, the mode of binding is not conserved. This indicates that heparin/HS binding of TcAs appears in a non-specific manner to concentrate Tc toxins on the host cell surface as a first step, and additional steps of receptor binding subsequently follow. The same binding model has been described for glycosaminoglycan binding of herpes virus¹⁰.

We added an additional chapter to the results section describing the structure of Xn-XptA1 (1.224ff). In addition, we rewrote the corresponding parts of the abstract (1.25ff), the introduction (1.77f), and the discussion (1.271ff) to compare both TcA structures with heparin.

Given these reservations about the heparin binding, I am somewhat skeptical about the proposed two-step binding model. In particular, the idea that heparin binding is the initial contact with a sub-nM KD seems at odds with the idea that the μ M binding of the Lewis antigens allows laterally mobile, dynamic binding of TcA to the cell surface.

We agree with reviewer 2 that the combination of both binding events will surely increase the overall affinity of TcA to the cell surface and therefore hinder potential lateral mobility. In addition, the weak and dynamic interaction between Lewis X and PI-TcdA1 is probably

enhanced when the correct protein moiety is additionally present in the real receptor. We therefore removed our speculative statements from the results (l.132ff) and discussion (l.289ff) and rewrote it accordingly.

Finally, it is not made clear if the observed glycan binding patterns are congruent with, and make a rational explanation for, the known cell tropism of different ABC toxins for different eukaryotic cell types.

Since various glycans are present on all insect and mammalian cells, our results explain why insecticidal ABC toxins can, to some extent, also infect mammalian cells. To explain the targeting of certain cell types with high specificity, the identification of the peptide moiety of the glycoprotein receptors is still missing. Moreover, the low affinity of the measured interaction contradicts the fact that sub-nanomolar toxin concentrations already kill the cells. Therefore, additional binding to the peptide portion of the receptor is most likely necessary to achieve a sub-nanomolar affinity (see also the point above), and points to a potential function of the additional RBDs.

We added these two points to the discussion (l.300ff and l.333ff, respectively):

“Conversely, the conserved Lewis X core structure is one of the receptor motifs that explains the toxicity of Pl-TcdA1 against various mammalian cells, although insects of the order *Lepidoptera* are the naturally targeted organisms ”

and

“However, the decreased toxicity seen in *N*-linked glycan deficient cells suggests that only synergistic binding will provide the affinity needed to infect cells with high specificity at sub-nanomolar Tc toxin concentrations.”

Shaun Lott, University of Auckland.

In addition to the reviewers' suggestions, we have changed the coloring of Supplementary Figures 3e and 6e to be accessible for colorblind readers. Furthermore, we have added an additional chapter to the Methods section that describes atomic model building and refinement.

References:

1. Heim, J. B., Hodnik, V., Heggelund, J. E., Anderluh, G. & Krenzel, U. Crystal structures of cholera toxin in complex with fucosylated receptors point to importance of secondary binding site. *Sci Rep* **9**, 12243–14 (2019).
2. Lang, A. E. *et al.* Phototaxin ADP-ribosylates actin and RhoA to force actin clustering. *Science* **327**, 1139–1142 (2010).
3. Meusch, D. *et al.* Mechanism of Tc toxin action revealed in molecular detail. *Nature* **508**, 61–65 (2014).
4. Gatsogiannis, C. *et al.* Tc toxin activation requires unfolding and refolding of a β -propeller. *Nature* **563**, 209–213 (2018).
5. Shukla, D. *et al.* A novel role for 3-O-sulfated heparan sulfate in herpes simplex virus 1 entry. *Cell* **99**, 13–22 (1999).
6. Artpradit, C. *et al.* Recognition of heparan sulfate by clinical strains of dengue virus serotype 1 using recombinant subviral particles. *Virus Res.* **176**, 69–77 (2013).
7. Mietzsch, M., Broecker, F., Reinhardt, A., Seeberger, P. H. & Heilbronn, R. Differential adeno-associated virus serotype-specific interaction patterns with synthetic heparins and other glycans. *J. Virol.* **88**, 2991–3003 (2014).
8. Piper, S. J. *et al.* Cryo-EM structures of the pore-forming A subunit from the *Yersinia entomophaga* ABC toxin. *Nat Commun* **10**, 1952 (2019).
9. Leidreiter, F. *et al.* Common architecture of Tc toxins from human and insect pathogenic bacteria. *Sci Adv* **5**, eaax6497 (2019).
10. Bartlett, A. H. & Park, P. W. Proteoglycans in host-pathogen interactions: molecular mechanisms and therapeutic implications. *Expert Rev Mol Med* **12**, e5 (2010).

REVIEWERS' COMMENTS:

Reviewer #2 (Remarks to the Author):

The revised manuscript improves in a number of aspects on the original version.

The logic behind the decision to pursue heparin binding is now more clearly described. I think the paper would be still be clearer if there was a single, coherent numbering scheme in the compounds used, rather than having to cross-reference different numbering schemes in supplementary Tables. I understand that renumbering the compounds for the paper would be a frustrating exercise, but I think it would improve the clarity of the paper for the reader.

The visualisation of heparin binding in two different TcA subunits certainly bolsters the claim that it is a mechanism of initial cell surface recognition in some (but not all) members of the TcA family, and the lack of conservation of the binding site implies that the recognition of heparin has evolved independently at least twice in the TcA family. This itself is an intriguing new finding. I still think the heparin screening experiment would be improved by the addition of: 1) a known heparin-binding protein as a positive control, and 2) another negative control that contains a positively charged surface patch. These controls would give information about the specificity of binding, but I accept that they would involve a reasonable amount of extra work and would do little to change the overall conclusions of the paper, so are not strictly required.

We thank reviewer 2 for the additional feedback. Below we include our detailed response to each point.

Reviewers' comments:

Reviewer #2 (Remarks to the Author):

The revised manuscript improves in a number of aspects on the original version.

The logic behind the decision to pursue heparin binding is now more clearly described. I think the paper would be still be clearer if there was a single, coherent numbering scheme in the compounds used, rather than having to cross-reference different numbering schemes in supplementary Tables. I understand that renumbering the compounds for the paper would be a frustrating exercise, but I think it would improve the clarity of the paper for the reader.

We thank reviewer 2 for the suggestion. With the Glycan-IDs, the first column in Supplementary Tables 2 and 3, there is already a consistent numbering scheme. We decided for the present numbering schemes in the figures to transmit their contents as clearly as possible. Therefore, we also included schemes of the relevant glycans whose interactions with TcAs are described in more detail in Figures 2 and 3, respectively. Re-numbering the glycans in the figure panels would be less understandable for a broad audience without a comprehensive expertise in glycan chemistry.

The visualisation of heparin binding in two different TcA subunits certainly bolsters the claim that it is a mechanism of initial cell surface recognition in some (but not all) members of the TcA family, and the lack of conservation of the binding site implies that the recognition of heparin has evolved independently at least twice in the TcA family. This itself is an intriguing new finding. I still think the heparin screening experiment would be improved by the addition of: 1) a known heparin-binding protein as a positive control, and 2) another negative control that contains a positively charged surface patch. These controls would give information about the specificity of binding, but I accept that they would involve a reasonable amount of extra

work and would do little to change the overall conclusions of the paper, so are not strictly required.

We agree with reviewer 2 that the addition of the suggested additional positive and negative controls would indeed improve the heparin screening assay. However, also without the suggested controls it is obvious that the binding signals of all four toxins are higher than those of the two negative control proteins WGA and BSA. We refer to adenovirus-2 that had been screened on the same array layout and shows binding signals in the same range as Mm-TcdA4 and Xn-XptA1 (l. 187ff in the manuscript). In addition, a biolayer interferometry experiment of Mm-TcdA4 and heparin suggests an affinity in the sub-micromolar range. Finally, our two structures of Mm-TcdA4/heparin and Xn-XptA1/heparin show the association of heparin to the toxins. In the case of Xn-XptA1, the interaction site is less dominated by positively charged residues like in the case of Mm-TcdA4. Therefore, the suggested additional negative control with a large positive patch is obsolete.

Altogether, we also agree with reviewer 2 that the additional information gained by including the two controls would not change the paper's conclusion, especially under the viewpoint of the results shown by structural biology. However, for future experiments we will consider the present suggestions.